

# A deep learning method for the recognition of solar radio burst spectrum

Jun-Cheng Guo[1], Fa-Bao Yan[2], Gang Wan[1], Xin-Jie Hu[1] and Shuai Wang[1]

[1] Space Engineering University, Beijing, China
[2] Laboratory for ElectromAgnetic Detection (LEAD), Insitute of Space Sciences, Shandong University, Weihai, Shandong, China

## ABSTRACT

Solar radiation is the excitation source that affects the weather in the atmosphere of the earth, and some solar activities such as flares and coronal mass ejections are often accompanied by radio bursts. The spectrum of solar radio bursts is helpful for astronomers to explore the mechanism of radio bursts. With the development and progress of solar radio spectrum observation methods, the observation of the Sun can be done at almost all times of day. How to quickly and automatically identify the small proportion of burst data from the huge *corpus* of observation data has become an important research direction. The innovation of this study is to enhance the original radio spectrum dataset with unbalanced sample distribution, and a neural network model for solar radio spectrum image classification is proposed on this basis. This hybrid structure of joint convolution and a memory unit overcomes the shortcoming of the traditional convolution or memory model, which can only extract one-sided features of an image. By extracting the frequency structure features and time-series features at the same time, the sensitivity to the small features of the spectrum image can be enhanced. Based on the data of the Solar Broadband Radio Spectrometer (SBRS) in China, the proposed network model can improve the average classification accuracy of the spectrum image to 98.73%, which will be helpful for related astronomical research.

## INTRODUCTION

Solar radio bursts originate from energetic electrons, which are radio wave signals. The frequency ranges from KHz to sub THz, and the intensity can increase to thousands to tens of thousands of times of the quiet sun intensity in a short time (*Du et al., 2017*). These high-intensity signals can enter our radio communication system and seriously interfere with our communication (*Tan, 2007*). The duration of radiation enhancement is from one second to several days. Solar radio is usually observed by dynamic spectrum instrument, and the data given are the spectrum of radio intensity changing with frequency and time, which is called the dynamic spectrum (*Xu et al., 2020*).

The Solar Broadband Radio Spectrometer (SBRS) in China (*Fu et al., 2004*) was put into operation during the 23rd solar cycle. By installing component spectrometers on different antennas in Beijing, Kunming and Nanjing, the SBRS can observe the sun in

Corresponding authors
Fa-Bao Yan, hjc-8555@sdu.edu.cn
Shuai Wang,
mage1120@foxmail.com

all-weather conditions, which also produces a large number of observation datasets, and there are very few data including solar radio burst events. At the same time, due to the influence of radio frequency interference (RFI) and other factors of the observation equipment, the original data cannot be quickly distinguished from the burst data and other types of data, which makes it difficult for subsequent astronomical research (*Yan et al., 2017*). Therefore, how to automatically process and classify the data observed by the SBRS will be of great help to the study of solar radio bursts.

With the development of computer performance and artificial intelligence technology, increasingly more deep learning models and algorithms are used to solve natural language processing, computer vision and speech recognition tasks (*Fu et al., 1995*). In light of the related problems in the field of astronomy, many previous works have used image processing algorithms and neural network models for reference, as well as big data technology and artificial intelligence methods to carry out the research of burst classification based on SBRS data, which has gradually been widely recognized (*Yan et al., 2020*). In the early stage of research, researches first used a Canny operator to detect and reduce the bad data of instrument or environmental interference, and then used Hough and Radon transforms to detect the approximate straight line in the spectrum, through line matching to analyze and identify type II and type III bursts. Thus, they realized the automatic detection of bursts (*Lobzin et al., 2009*; *Lobzin et al., 2010*). Although this method based on statistical characteristics is efficient, the calculation parameters must be designed artificially, and the experimental results are not universal. In recent years, as convolution neural networks (CNNs) (*Krizhevsky, Sutskever & Hinton, 2012*), long short-term memory network (LSTM) (*Hochreiter & Schmidhuber, 1997*), and deep belief networks (DBNs) (*Bengio, 2009*) have become increasingly more mature, many researches begin to apply them to solve the classification problem of solar radio spectra (*Xu et al., 2003*; *Yang, Wang & Hu, 2019*). In February 2016, *Chen et al. (2016)* established a solar radio spectrum image and its label database based on SBRS data, and used deep learning method to solve the problem of spectrum image classification for the first time, which set off an upsurge of solar radio spectrum automatic classification research. From DBNs in 2016 to multimodal deep learning (MDL) in 2017 to CNNs in 2018 and LSTM in 2019 (*Ma et al., 2017*; *Chen, 2018*; *Xu et al., 2019*), researches have been continuously improving the classification accuracy of dynamic spectra by designing more ingenious neural network structures. However, with the improvement of the model structure, the improvement effect on the recognition accuracy is becoming increasingly smaller. In May 2019, without changing the structure of the classifier, the researcher expanded the samples in the database by using the random index of samples, and then classified the spectrum images with a CNN (*Zheng, 2019*). In addition, some previous works have proposed automatic identification methods for specific types of solar radio burst structures (*Salmane et al., 2018*; *Lobzin et al., 2010*; *Zhang, Wang & Ye, 2018*; *Dayal et al., 2019*).

In 2020 (*Guo et al., 2020*), we improved the process of spectrum image processing, added a threshold segmentation operation to make the input CNN image features more obvious, and used a random index to expand the number of samples. In this study,

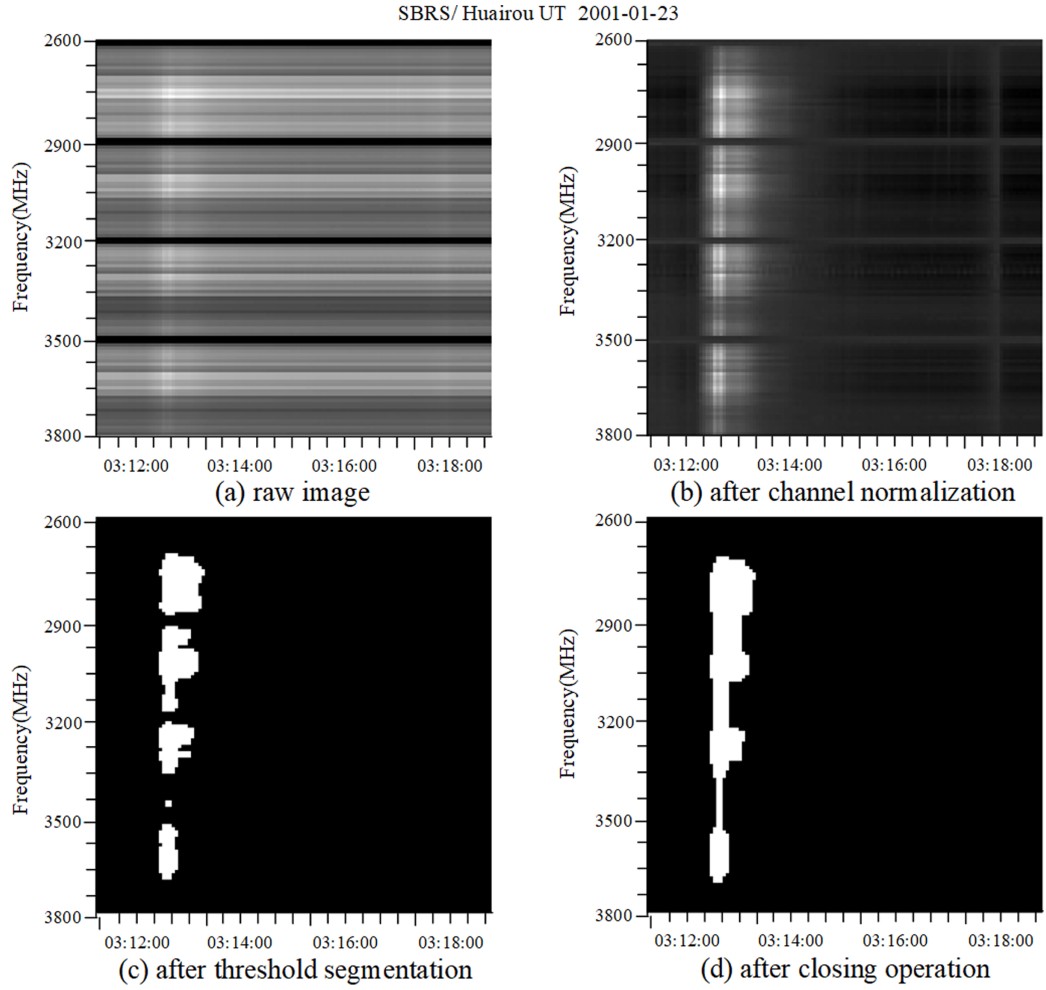

**Figure 1** **Dynamic spectra of burst after pre-processing.** This figure represents the pixel value of the spectrum image on a certain time channel $x$ and frequency channel $y$.

we continue to use threshold segmentation for image pre-processing, but with the following improvements:

1. A hybrid model of a convolution and memory unit is proposed as the classifier of spectrum images, which makes it more sensitive to the small features of spectrum images by extracting the frequency structure features and temporal sequence features of such images at the same time.

2. An affine transform combined with random selection is used to improve the classification accuracy, and the effective number of some types of samples is expanded.

## PRE-PROCESSING OF DYNAMIC SPECTRA

### Channel normalization

It can be seen from Fig. 1A that, due to the influence of the channel effect of observation equipment, the appearance of interference signal noise conceals the real

characteristics of the solar radio burst, which makes the dynamic spectra features not obvious enough. To reduce the influence of RFI on the spectrum image, we use the channel normalization method to remove the RFI, and the calculation method is as follows:

$$p^{'}(x,y) = p(x,y) - \frac{1}{n}\sum_{y=0}^{n} p(x,y) + \frac{1}{mn}\sum_{x=0}^{m}\sum_{y=0}^{n} p(x,y), \tag{1}$$

where $p(x,y)$ represents the pixel value of the spectrum image on a certain time channel $x$ and frequency channel $y$, $p^{'}(x,y)$ is the pixel value after channel normalization. The radio spectrum image after the horizontal stripes are reduced is shown in Fig. 1B. It can be seen that the spectrum image at this time is free of the influence of RFI, and its burst characteristics gradually appear.

## Threshold segmentation

The purpose of channel normalization is to reduce the difficulty of dynamic spectra feature recognition caused by instrument noise. However, both burst type and other types of dynamic spectra images have the problem of over-fusion of image backgrounds and features. To further enhance the dynamic spectra features, we use threshold segmentation algorithm to binarize the dynamic spectra. According to the gray value range of spectrum images of from the SBRS, the threshold segmentation algorithm is designed as follows:

$$p^{'}(x,y) = \begin{cases} 255, p(x,y) > (mod(p) + max(p))/2 \\ 0, \quad p(x,y) \le (mod(p) + max(p))/2 \end{cases}, \tag{2}$$

where $mod(p)$ is the gray value of the maximum value of the spectrum image gray histogram, and $max(p)$ is the maximum value of spectrum image gray. Gray histogram counts the frequency of each gray value in the image, $mod(p)$ can be calculated by first counting the occurrence times of the gray value of the whole spectrum image to obtain the gray histogram, and then finding the gray value of the image with the most occurrence times in the gray histogram, $max(p)$ is the maximum gray value of the whole spectrum image, which can be calculated by sorting all gray values in the image. In particular, they are all computed for the entire dynamic spectrum. The binarization of spectrum images by a threshold segmentation algorithm is to segment the whole image into regions with only 0 and 255 gray levels according to the burst characteristics of the spectrum image and the different gray levels occupied by a quiet background. The processing result is shown in Fig. 1C. It can be seen that the dynamic spectra features are discontinuous bright spots distributed along the frequency channel as a whole.

## Morphological closed operation

A morphological operation is an image processing method based on set theory of data morphology for binary image. The most basic morphological operations include dilation and erosion. The mathematical definition of expansion and erosion of image $A$ by structural element $B$ are as follows:

$$A \oplus B = \left\{ z | (B)_z \cap A \neq \varnothing \right\} \tag{3}$$

$$A \Theta B = \left\{ z | (B)_z \subseteq A \right\} \tag{4}$$

A morphological close operation is used to expand the image first and then corrode it. It can filter the image by filling the concave angle of the image. It can be used to fill the small holes in the image, bridge the small cracks, and keep the total position and shape unchanged (*Serra, 1982*). The spectrum image after a morphological close operation is shown in Fig. 1D, from which we can see that the close operation repairs the intermittent characteristics of the burst type spectrum image, so that the burst type spectrum image basically contains obvious bright spot vertical stripes. Similarly, the calibration type spectrum image contains large bright and dark areas after morphological close operation; the non-burst type spectrum image is basically comprised of dark areas, but some images also exhibit white noise.

## CGRU NEURAL NETWORK

CNN is a type of feedforward neural network that includes convolution operations and has a deep structure, which is built to mimic the visual perception of living beings. A CNN is mainly composed of an input layer, convolutional layer, pooling layer, fully connected layer and output layer (*He & Gong, 2017*). Convolutional layer performs feature extraction on the input image, and it contains multiple convolution kernels. Pooling layer is mainly used for down-sampling dimension reduction, removing redundant information, compressing features, simplifying network complexity, and reducing computation and memory consumption. The fully connected layer is to non-linearly combine the features extracted by the convolutional layer and the pooling layer, and then obtain the output through the activation function (*Krizhevsky, Sutskever & Hinton, 2012*).

The traditional neural network only relies on the current input and is not sensitive to location information, which makes it unable to perceive the dependence between time series data. A recurrent neural network (RNN) solves the problem of variable-length input and timing dependence by connecting feedback in the hidden layer, so it can memorize the input information (*Lample et al., 2016*). The gated recurrent unit (GRU) is a special type of RNN. By introducing a controllable self-loop, a GRU constructs a controllable memory unit containing two gate-control switches, which solves the problem that a RNN can only learn short-term features due to gradient explosion and gradient disappearance. (*Chung et al., 2014*; *Kyunghyun et al., 2014*). The controllable memory unit structure of a GRU is shown in Fig. 2.

For the solar radio spectrum image, we proposes a hybrid network structure combining convolution and gated memory units, which is called CGRU. The network structure is shown in Fig. 3, and the structural parameters are shown in Table 1. The input of this model is the pre-processed dynamic spectra, and the output labels are divided into three categories, namely, Non-burst, Burst, and Calibration types, which are represented by 0, 1, and 2, respectively. The whole CGRU network is composed of convolution and memory part. The convolution part includes three convolution layers and three max
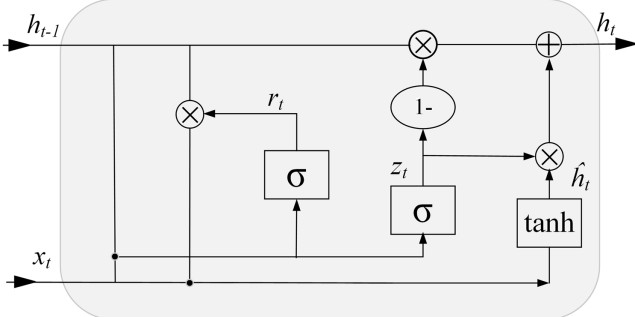

**Figure 2  GRU memory unit structure.**     

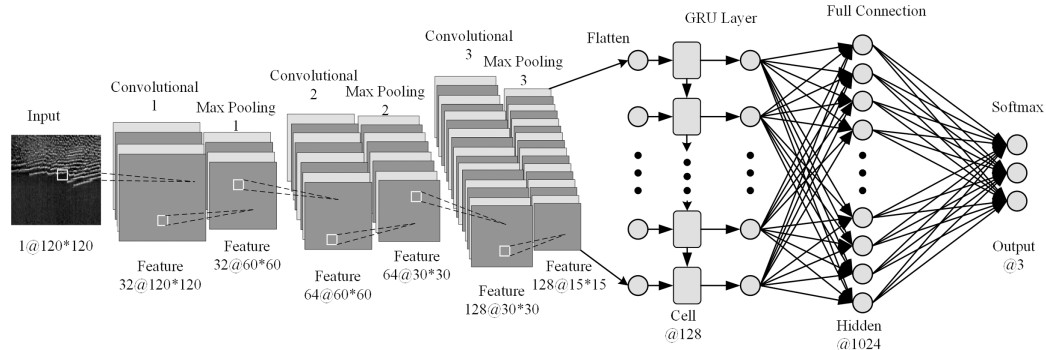

**Figure 3  CGRU neural network structure.**     

| Table 1  Number structural parameters of CGRU model. | | | | |
|---|---|---|---|---|
| **Layer (type)** | **Kernel size** | **Stride** | **Parameters** | **Output shape** |
| Input | N/A | N/A | N/A | (120, 120, 1) |
| Convolution1D | 5 | 1 | 19,232 | (120, 32) |
| Maxpooling1D | 2 | 2 | 0 | (60, 32) |
| Convolution1D | 5 | 1 | 10,304 | (60, 64) |
| Maxpooling1D | 2 | 2 | 0 | (30, 64) |
| Convolution1D | 5 | 1 | 24,704 | (30, 128) |
| Maxpooling1D | 2 | 2 | 0 | (15, 128) |
| GRU | N/A | N/A | 98,688 | 128 |
| Full connection | N/A | N/A | 132,096 | 1,024 |
| Output(Softmax) | N/A | N/A | 3,075 | 3 |

pooling layers. The convolution layer (Conv1D) can capture the burst information of the spectrum image and extract its spatial features, and multiple convolution layers can improve the ability of the model to extract image spatial features and improve the recognition accuracy of the model. However, with the increase of the number of convolution layers, the accuracy of the model will tend to be stable, it will even lead to over fitting phenomenon; the pooling layer (Maxpooling1D) is used to reduce the feature dimension and reduce the model burden. The memory part is composed of GRU, which

| Table 2 Statistics on solar radio spectrum database. | | | | |
|---|---|---|---|---|
| Type | Non-burst | Burst | Calibration | Total |
| Number | 3,335 | 579 | 494 | 4,408 |
| Size | 240 × 2,520 | 240 × 2,520 | 240 × 2,520 | 240 × 2,520 |

can extract the time-series features of the image. The output of GRU needs to be flattened first, then through the full connection layer, and finally transfer the feature vector to the softmax layer to get the final classification result. The bidirectional GRU (BiGRU) combines the forward and backward features of the input sequence on the basis of the GRU, and can capture the context information of the input sequence, which is meaningful for the model to improve the feature extraction ability. The hybrid network structure of joint convolution and bidirectional gated memory unit is called CBiGRU.

## EXPERIMENTAL RESULTS

### Solar radio spectral data enhancement

In 2016, *Chen et al. (2016)* established the solar radio spectrum database, and the spectrum image data are from the SBRS. The database collected 4,408 radio spectrum images with a size of 2,520 × 240. These images are divided into three categories: burst for radio spectrum images with burst behavior, non-burst for images without burst behavior, and calibration for images requiring secondary calibration. The distribution of various types of samples in the original dataset is shown in Table 2. It can be seen that the number of non-burst images accounts for 76% of the total number of samples, while the number of the other two types of samples is relatively small. This unbalanced sample distribution will greatly limit the advantages of deep learning, and it will not be able to make full use of all radio data to train the network, which will easily cause the network model to overfit and adversely affect the final experimental results. In previous studies, except for the random index method used by *Zheng (2019)* to enhance the database samples, other researches have improved the classification accuracy by improving the classification model. However, the random index only adds the existing images to all kinds of samples randomly, which cannot effectively improve the sample richness of the dataset. Therefore, this paper proposes a data enhancement method based on an affine transform and random index to expand the solar radio spectral dataset, improve the sample richness, and solve the problems of balanced sample distribution and sample data in the dataset.

First, the original data are divided into left and right circular polarization parts according to single and double channels. The size of a single image is scaled to 120 × 2,520. At the same time, the data of all kinds of samples are doubled, and the total number of samples is 8,816. Secondly, due to the imbalance of burst and calibration type samples, affine transformation is used for data enhancement. The specific operation is to first flip the original image horizontally, change the temporal characteristics of the spectrum image, and double the number of samples; the original image is flipped vertically to change its frequency characteristics. After two image flipping operations, the number of burst and calibration type samples reaches 3,474 and 2,964 respectively. Finally, to further

**Table 3 Number of entries after sample enhancement.**

| Type | Non-burst | Burst | Calibration | Total |
|---|---|---|---|---|
| Number | 6,670 | 6,670 | 6,670 | 20,010 |
| Size | 120 × 2,520 | 120 × 2,520 | 120 × 2,520 | 120 × 2,520 |

compensate for the gap in the number of samples with the non-burst type, the random index method is used to increase the number of various samples to 6,670. At this time, the dataset contains 20,010 dynamic spectra, which can basically meet the data requirements of model training. Table 3 shows the sample distribution of the solar radio spectrum dataset after data enhancement.

## Evaluation criterion

We use true positive rate (TPR) and false positive rate (FPR) to evaluate the experimental effect, which is calculated as follows:

$$TPR = \frac{TP}{TP + FN} \times 100\%, \tag{5}$$

$$FPR = \frac{FP}{FP + TN} \times 100\%. \tag{6}$$

where $TP$ represents a correctly classified positive sample, $FN$ represents a wrongly classified positive sample; $TN$ represents a correctly classified negative sample, and $FP$ represents a wrongly classified negative sample. TPR can be expressed as the proportion of positive samples that are correctly predicted. Its numerator is the number of positive samples that are correctly classified and the denominator is the number of all positive samples. FPR represents the proportion of negative instances that are predicted; its numerator is the number of misclassified negative samples, and its denominator is the number of all negative samples. Generally speaking, if the TPR value of a certain type of sample is larger, it means that the probability of successful identification of this type is greater. If the FPR value is larger, it means that the probability of other class samples being incorrectly identified as this type is greater.

## Performance comparisons

Before inputting all the data into the network, we preprocessed the spectrum images in the dataset. First, the image standardization method is used to rescale the gray value to the range of 0–255; second, in order to speed up the network training speed, we limit the size of the image to 120 × 120, Fig. 4 shows the spectrum image changes before and after downsampling, it can be seen from its gray-scale histogram that the down-sampling operation will not significantly change its statistical characteristics; finally, the image is filtered and feature-enhanced using methods such as channel normalization, threshold segmentation, and morphological operations. In addition, we label the non-burst type image in the dataset as [1 0 0], the burst type data as [0 1 0], and the calibration type data as [0 0 1]. At the same time, we divide the dataset into trainingset and testingset according to the ratio of 4:1, including 16,008 training pictures and 4,002 testing pictures.

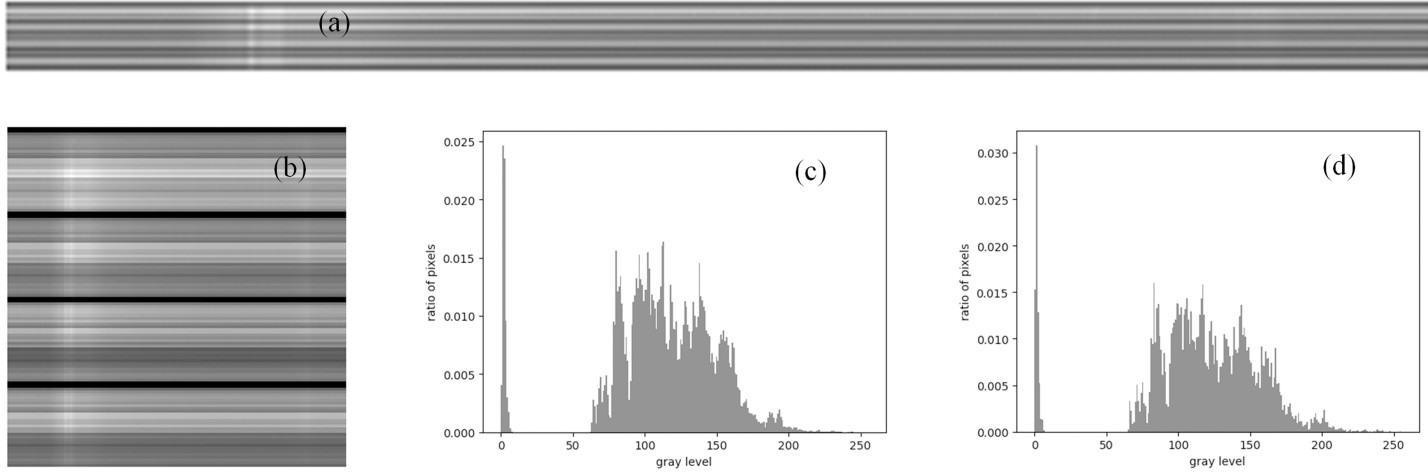

**Figure 4 Solar radio spectrums and gray histograms before and after downsampling observed by the SBRS.** (A) Original solar radio burst spectrum image, (B) enlarged downsampled solar radio spectrum image, (C) histogram before downsampling, and (D) histogram after downsampling.

**Table 4 The hardware configuration and processing speed used in the experiment.**

| GPU | CPU | Base board | Physical memory | Processing speed | Training speed |
|---|---|---|---|---|---|
| NVIDIA TITAN RTX 24G | Intel i9-9900k | Intel Z390 UD | 64 G | 520 μs/image | 6.3 s/epoch |

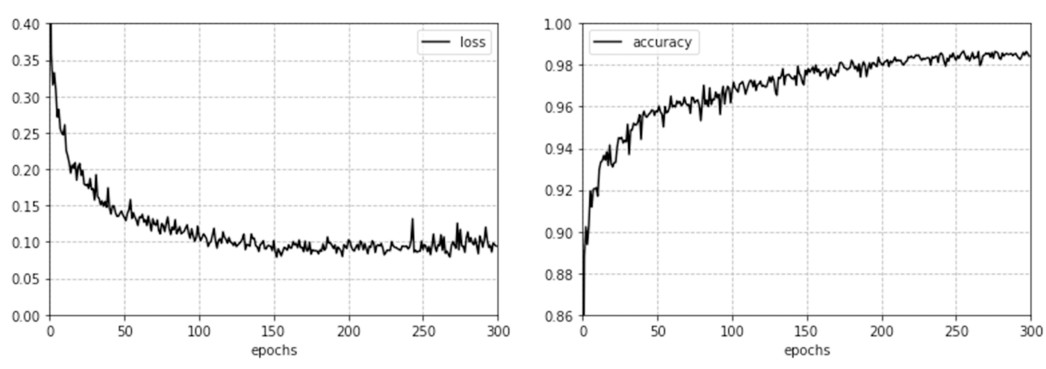

**Figure 5 Loss and accuracy curve of the CGRU model.**

To reflect the influence of different pre-processing methods and network structures on the recognition accuracy and verify the effectiveness of the method advanced in this paper, two comparative experiments were designed. The adam optimizer was used, the number of iterations was 300, the size of the batch was 128, the learning rate was 0.0001, the time-series length processed by the memory unit was 120, and the number of hidden neurons was maintained at 128. The hardware configuration and processing speed used in the experiment are listed in Table 4. Figure 5 shows the change of model loss and validity accuracy during model training.

**Table 5 Effect of different convolutional layers.**

| TPR (%) | | Number of convolutional layers | | | |
|---|---|---|---|---|---|
| | | 3 | 4 | 5 | 6 |
| CGRU | Non-burst | 96.55 | 96.78 | 96.03 | N/A |
| | Burst | 99.55 | 99.48 | 99.63 | N/A |
| | Calibration | 99.92 | 99.93 | 100.0 | N/A |
| CBiGRU | Non-burst | 95.88 | 95.13 | 96.40 | 95.58 |
| | Burst | 100.0 | 100.0 | 99.93 | 100.0 |
| | Calibration | 100.0 | 100.0 | 100.0 | 100.0 |

**Table 6 Performance comparisons.**

| Reference | Model | Data enhancement | | | Non-burst | Burst | Calibration | Accuracy |
|---|---|---|---|---|---|---|---|---|
| Chen et al. (2016) | Deep belief nets (DBNs) | NO | | TPR | 86.4 | 67.4 | 95.7 | 83.17 |
| | | | | FPR | 14.1 | 3.2 | 0.4 | 5.90 |
| Ma et al. (2017) | Multimodal deep learning (MDL) | NO | | TPR | 83.3 | 82.2 | 92.5 | 86.00 |
| | | | | FPR | 13.9 | 15.6 | 3.2 | 10.90 |
| Chen (2018) | CNN | NO | | TPR | 90 | 84.6 | 100 | 91.53 |
| | | | | FPR | 10 | 8.6 | 0.3 | 6.30 |
| Xu et al. (2019) | LSTM | NO | | TPR | 92.3 | 85.4 | 96.2 | 91.30 |
| | | | | FPR | 8.2 | 6.7 | 0.9 | 5.27 |
| Zheng (2019) | CNN | Random index | | TPR | 99.3 | 98.4 | 85.6 | 94.43 |
| | | | | FPR | 1.4 | 0.8 | 6.2 | 2.80 |
| Guo et al. (2020) | CNN | Random index | | TPR | 93.9 | 100 | 100 | 97.97 |
| | | | | FPR | 0.0 | 2.7 | 0.4 | 1.03 |
| Present paper | CGRU | Random index + Affine | | TPR | 96.8 | 99.5 | 99.9 | 98.73 |
| | | | | FPR | 0.0 | 1.5 | 0.3 | 0.6 |

Table 5 shows the effect of different convolutional layers on the recognition accuracy of CGRU and CBiGRU. It can be seen that the deepening of convolution layers has a very limited effect on the mining of a spectrum image features after pre-processing. This is because too many convolution layers will cause the network to over fit and ignore the small features of the image, resulting in the reduction of the recognition accuracy of the two models. Although BiGRU adopts a two-way network structure, it is different from natural language processing tasks in that spectrum images do not need to extract contextual features. For time-series images, only historical information must be considered, so the overall recognition accuracy is almost the same as that of the GRU.

Table 6 shows the results of comparison of the experimental results of all the methods from the literature. The following can be found from the table.

1. The calibration type spectrum image has relatively simple characteristics, so basically all models have achieved relatively high recognition accuracy. The TPR value obtained in

*Zheng (2019)* is significantly lower than that of other models. This is because the image pre-processing method is too simple and causes the model to fail to distinguish the image background and features.

2. The recognition accuracy of the burst and non-burst types in *Zheng (2019)* is significantly higher than that of other models. The reason is that it expands the original sample to solve the problem of sample imbalance, but the classification model it uses has not changed much.

Compared with the classification results in the existing literature, the recognition accuracy of the proposed method is improved by 0.76%, and the optimal result is 98.73%. The TPR value of non-burst type is close to 96.8%, which is slightly lower than the best result. By comparing the experimental methods and results, the following conclusions can be drawn.

1. By reasonably expanding the sample dataset and increasing the effective number of various samples, the balance of neural network training and testing can be improved, and the classification accuracy can be improved.
2. Because the background noise of the original dynamic spectra is relatively strong, it is difficult for the neural network to directly extract the image features. The feature enhancement of the sample before input to the neural network is helpful to improve the final recognition effect.
3. The neural network structure of CGRU can extract the image features. Spatial features can capture time-series features and are more sensitive to solar radio bursts.
4. In the process of feature enhancement in this paper, although the feature of non-burst dynamic spectra is enhanced after closed operation, it also samplifies the background noise in the original spectrum image, which is the main reason that causes non-burst type to be recognized as burst type.

## CONCLUSION AND DISCUSSION

The study of solar radio bursts is helpful to improve the understanding of the physical mechanism of solar activity. Long-term observations of the sun have brought massive amounts of radio spectrum data. However, these data cannot be directly used for analysis and research, and classification and archiving are required. Subject to the influence of the received image quality and data volume, the effectiveness and timeliness of traditional classification methods can no longer meet the task requirements. Based on this, we apply deep learning technology and image processing algorithms to the classification of spectrum images, mainly improving the quality of spectrum images, increasing the number of some types of spectrum images, and designing a spectrum image classification network.

Through our experimental results, we have got some conclusions: the original image is enhanced to reduce the interference of background noise; by combining affine transformation and a random index to expand the database, the problem of unbalanced sample distribution is solved; the convolution structure and memory structure are used to

capture the spatial and temporal characteristics of the spectrum image to improve the ability of neural network to detect the burst behavior, and finally increase the classification accuracy of the solar radio spectrum image to 98.73%, which is the best classification result compared with historical literatures. At the same time, the current model also has some problems, which we hope to improve in future work: down-sampling the original image will cause damage to the burst morphological structure of the radio spectrum, which will affect the recognition accuracy; the threshold of the morphological closed operation is difficult to grasp, unreasonable settings will amplify the background noise in the original spectrum image, leading to errors in classification.

In fact, according to the morphology of the radio bursts in dynamic spectrum, radio bursts in the meter wave band can be further divided into I–V types (*Lobzin, Cairns & Zaslavsky, 2014*). Each type of radio burst is closely related to different solar activity phenomena, which is an important means to understand and cognition the mechanism of solar burst. In future work, we will carry out research on the classification of different types of radio bursts to achieve automatic identification of the entire process from screening burst images to detecting burst types and positions.

### Funding

This work is supported by the National Natural Science Foundation of China (Grant Nos. 41904158 and 11703017), The China. Postdoctoral Science Foundation (Grant No. 2019M652385), and the Young Scholars Program of Shandong University, Weihai (Grant No. 20820201005). There was no additional external funding received for this study. The funders had no role in study design, data collection and analysis, decision to publish, or preparation of the manuscript.

### Grant Disclosures

The following grant information was disclosed by the authors:
National Natural Science Foundation of China: 41904158, 11703017.
China Postdoctoral Science Foundation: 2019M652385.
Young Scholars Program of Shandong University, Weihai: 20820201005.

### Competing Interests

The authors declare that they have no competing interests.

### Author Contributions

- Jun-Cheng Guo conceived and designed the experiments, performed the experiments, performed the computation work, prepared figures and/or tables, and approved the final draft.
- Fa-Bao Yan conceived and designed the experiments, analyzed the data, prepared figures and/or tables, and approved the final draft.
- Gang Wan analyzed the data, performed the computation work, prepared figures and/or tables, authored or reviewed drafts of the paper, and approved the final draft.

- Xin-Jie Hu analyzed the data, prepared figures and/or tables, authored or reviewed drafts of the paper, and approved the final draft.
- Shuai Wang performed the experiments, authored or reviewed drafts of the paper, and approved the final draft.

## Data Availability

The data is available at GitHub: https://github.com/filterbank/spectrumcls.

## Supplemental Information

Supplemental information for this article can be found online at http://dx.doi.org/10.7717/peerj-cs.855#supplemental-information.

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
