# Peer review of "A deep learning method for the recognition of solar radio burst spectrum"

_PeerJ Computer Science, doi:10.7717/peerj-cs.855_

## Round 0.1 · original submission · Major Revisions

The authors should address review comments to revise the manuscript.

·

Basic reporting

Line 31: Solar radio burst is (not originates from) radio wave signal. Solar radio bursts originate from energetic electrons.
Line 32: Unit of frequency should be Hz. Meter kilometer-wave is the wavelength.
Line 36: Height (or heliocentric distance) of solar radio bursts is not determined by propagation characteristics, it’s determined by emission mechanism and background plasma property.
Line 40: Description not correct, the author of “Tan et al 2007” is not “Australian astronomers”. And “Tan et al 2007” in the reference list doesn’t have referred journal volume information.
Line 62: [Scholars] -> [researches] or [previous works] or [works in literature]. (this problem happened multiple times in the paper, please correct them thorough)
Line 71: Citation form: Author-year
Line 92: gray-scale value after normalization could be out of the range [0,255], does here include change of data type (from uint8 to another type), or how to rebin the normalized value to the range of [0,255]
Line 115: Redundant description of a common method widely used in computer vision, please reduce the description length and cite the original paper of morph-close [Image Analysis and Mathematical Morphology by Jean Serra (1982)]
Line 140-202: Redundant description of CNN, please replace this part with some citation (e.g: ImageNet classification with deep convolutional neural networks 2012 [proceedings-neurips] A Krizhevsky et al) or any textbook introducing CNN, GRU is well described in the paper of [Cho, Kyunghyun, et al 2014]
Line 217: delete line break
Line 309-316: not the conclusion of this paper.

[For the basic concept of solar radio bursts, I highly recommend the author to read the book: Introduction to Solar Radio Astronomy and Radio Physics]

Experimental design

(1) I wouldn’t suggest the authors claim they proposed a “new” method or model, as the CNN-GRU network is already widely used, for example:
[Hybrid CNN-GRU model for high efficient handwritten digit recognition 2019 Vantruong Nguyen] and [Detecting Hate Speech on Twitter Using a Convolution-GRU Based Deep Neural Network]
(2) The description of the dataset is important, please elaborate on the data preparation procedure: how the data is labeled and segmented, input size of the network is 120*120 while the data size is 120*2520. Please state how 120*2520->120*120 is achieved, either one dynamic spectrum is divided into 21 standalone images or the image is rescaled? Downsampling rescale will cause the loss of resolution and the loss of fine structures. (This paper is about fine structures, hence should be sensitive to resolution, so I wouldn't suggest any sort of downsampling before the input of the network).
(3) The detail of affine transformation needs to be elaborated.
(4) Affine transformation may not be appropriate in the context of solar radio bursts, because the type of solar radio burst is determined by its shape and frequency drift rate, and affine transformation of the dataset may change the shape and frequency and create a dynamic spectrum that won’t exist in the real world. This will make the model perform largely differently in the authors’ dataset and fresh new data.

Validity of the findings

(1) In this work, the output information is limited: [Event/No-event/Calibration]. Namely the existence of solar radio bursts. The output doesn’t include the information of the type of solar radio bursts and the position in the dynamic spectrum. But actually, the information of [what type, when, and which frequency range] is more useful.

There are already methods that can obtain the type and time-frequency information of radio bursts:
https://www.frontiersin.org/articles/10.3389/fphy.2021.646556/full
https://www.swsc-journal.org/articles/swsc/pdf/2018/01/swsc170092.pdf
https://meetingorganizer.copernicus.org/EGU2020/EGU2020-5109.html

if the method can not be improved to output the type and time-frequency information, please at least discuss it in [future work]

(2) There should be a discussion session

(3) It would be better if the event list in this work can also be published

Reviewer 2 ·

Basic reporting

Comments to the Author

A paper documenting a deep learning technique for the fine structure recognition of solar radio spectrum burst is welcomed. Your manuscript contains much useful information, but I found some aspects requiring more thorough expressed. Here are my specific comments and suggestion:

1) The title may be propitiate to change to "solar radio burst spectrum".
2) Line 39-40, Line 309-316 have little relevance with the subject of this paper. I suggest removing them.
3) Line 192-193, eq(12)-eq(13) have bad display.
4) Line 216-217, Line 299-301 should be concatenated in one line.
5) Line 248, FN is defined twice.
6) Line 286, Line 289, it is hardly to see the where the Ref.12 locates in Table 6.
7) Some abbreviations should be fully explained when appearing in the first time. Such as CGRU, CBIGRU, TPR, FPR, TP, FN.
8) Line 319, what does the unbalanced sample distribution mean? It should be explained more thoroughly in advance.

Experimental design

No comments.

Validity of the findings

The author draws a conclusion that CGRU in this paper is more sensitive to small features of solar bursts than other methods. If it was drawn from Table 6, I suggest the author adds more support data such as the classification to complex features or structures in all kinds of solar bursts to make the conclusion solid.

Additional comments

No comments.

·

Basic reporting

The manuscript proposes a CGRU model for solar spectrum image analysis, especially for the recognition of burst radio spectrum. Owing to the preprocessing and GRU, experiments demonstrates that the proposed method is very efficient with high accuracy. Therefore, it gives new light on solar image understanding. However, it would be better to further explain the following:
1) Give the overview of the model architecture or/and the table of model hierarchy structure to clearly show that the model is much simple and efficiency;
2) Some typing errors need attentions, such the equation (12) and (13). The “FN” in line 249th seems be a mistake of “TN”.

Experimental design

no comment

Validity of the findings

no comment

Additional comments

no comment

---

## Round 0.2 · Minor Revisions

The authors should address review comments to revise the manuscript.

·

Basic reporting

Line 176, what’s the meaning of left right rotation, is it left and right polarization?

Figure 1, the y-axes should be frequency (MHz GHz), x-axes should be UT time.

Line 204, resolution should be MHz per pixel or sec per pixel, 120*120 is image size

Caption of Figure 4 is not complete.

Experimental design

False negative cases are important, please check case by case if there is event which is not marked in the original dataset.

Validity of the findings

The SBRS is open public:
https://nadc.china-vo.org/data/data/sbrs/
https://sun.bao.ac.cn/SHDA_data/
SBRS has no restrictions to publish data processing results, including the science results and event catalog. The event list is an important part of method validation.

Additional comments

The author has addressed some of the comments, I still have some concerns about the manuscript and results

Reviewer 2 ·

Basic reporting

I am glad to see the author gave quick responses and made appropriate revisions based on all reviewers' suggestions. The missing abbreviations and concepts are explained, and the wrong formats are fixed as well. The conclusions are re-organized and re-expressed so the tables and figures seem to justify them.

There are still some typo and small grammar errors in the document. I think they could be corrected through some automation tools before published. Anyway, the article is suggested to be accepted and published.

Experimental design

No comments.

Validity of the findings

No comments.

Additional comments

No comments.

---

## Round 0.3 · Minor Revisions

The authors should address review comments to revise the manuscript.

·

Basic reporting

The author has addressed most of my previous problems, these are some minor suggestions:

Line 30: kHz to sub THz
Line 31: background radiation -> quiet sun. background radiation usually refers to ‘The cosmic microwave background’ in astronomy
Line 40: channel noise -> RFI (radio frequency interference)
Line 92: eliminate -> reduce, instrument noise can never be perfectly removed.
Line 203: limit -> rescale or map.
Line 259: eliminate -> reduce, (this is used multiple times, please correct all)
Line 269: the radio signal changes on the dynamic spectrogram -> the morphology of the radio bursts in dynamic spectrum

I would suggest accepting the manuscript after minor revision.

Experimental design

no comment

Validity of the findings

no comment

---

## Round 0.4 · accepted · Accept

The reviewer has recommended accepting the manuscript.

·

Basic reporting

No further suggestions, the manuscript can be accepted

Experimental design

no comment

Validity of the findings

no comment